# Cognitive Function Is Associated with the Genetically Determined Efficiency of DNA Repair Mechanisms

**DOI:** 10.3390/genes15020153

**Published:** 2024-01-24

**Authors:** Nicolas Cherbuin, Hardip Patel, Erin I. Walsh, Ananthan Ambikairajah, Richard Burns, Anne Brüstle, Lene Juel Rasmussen

**Affiliations:** 1National Centre for Epidemiology and Population Health, Australian National University, Canberra 2601, Australia; erin.walsh@anu.edu.au (E.I.W.); ananthan.ambikairajah@canberra.edu.au (A.A.); richard.burns@anu.edu.au (R.B.); 2John Curtin School of Medical Research, Australian National University, Canberra 2601, Australia; hardip.patel@anu.edu.au (H.P.); anne.bruestle@anu.edu.au (A.B.); 3Discipline of Psychology, University of Canberra, Canberra 2617, Australia; 4Centre for Ageing Research and Translation, Faculty of Health, University of Canberra, Canberra 2617, Australia; 5Department of Cellular and Molecular Medicine, Center for Healthy Aging, University of Copenhagen, 2200 Copenhagen, Denmark; lenera@sund.ku.dk

**Keywords:** DNA repair, cognitive decline, brain ageing, oxidative stress, inflammation, single nucleotide polymorphism

## Abstract

Several modifiable risk factors for neurodegeneration and dementia have been identified, although individuals vary in their vulnerability despite a similar risk of exposure. This difference in vulnerability could be explained at least in part by the variability in DNA repair mechanisms’ efficiency between individuals. Therefore, the aim of this study was to test associations between documented, prevalent genetic variation (single nucleotide polymorphism, SNP) in DNA repair genes, cognitive function, and brain structure. Community-living participants (*n* = 488,159; 56.54 years (8.09); 54.2% female) taking part in the UK Biobank study and for whom cognitive and genetic measures were available were included. SNPs in base excision repair (BER) genes of the bifunctional DNA glycosylases *OGG1* (rs1052133, rs104893751), *NEIL1* (rs7402844, rs5745906), *NEIL2* (rs6601606), *NEIL3* (rs10013040, rs13112390, rs13112358, rs1395479), *MUTYH* (rs34612342, rs200165598), *NTHL1* (rs150766139, rs2516739) were considered. Cognitive measures included fluid intelligence, the symbol–digit matching task, visual matching, and trail-making. Hierarchical regression and latent class analyses were used to test the associations between SNPs and cognitive measures. Associations between SNPs and brain measures were also tested in a subset of 39,060 participants. Statistically significant associations with cognition were detected for 12 out of the 13 SNPs analyzed. The strongest effects amounted to a 1–6% difference in cognitive function detected for *NEIL1* (rs7402844), *NEIL2* (rs6601606), and *NTHL1* (rs2516739). Associations varied by age and sex, with stronger effects detected in middle-aged women. Weaker associations with brain measures were also detected. Variability in some BER genes is associated with cognitive function and brain structure and may explain variability in the risk for neurodegeneration and dementia.

## 1. Introduction

Cognitive decline and brain ageing are acute concerns in a global population predicted to age rapidly in the coming decades. Progressive changes in cognitive function and brain health are known to be associated with an increased risk of developing dementia later in life [1]. As dementia is a major cause of disease burden in developed countries and a looming problem in developing countries where life expectancy is quickly rising [2], factors that may accelerate or inhibit the development of cognitive decline have major social and economic implications. There is, therefore, a need to better understand the contributors to brain and cognitive ageing and to more precisely identify those at risk to better direct limited clinical and risk reduction interventions.

Healthy ageing and longevity in humans are modulated by a combination of genetic and non-genetic factors. To address these concerns, increased attention has been directed at modifiable risk factors (e.g., cardiovascular disease, diabetes, obesity, depression, sedentary lifestyle, unhealthy diet, smoking, etc.), which are thought to contribute significantly (~40%) to brain ageing and cognitive decline, as well as other chronic conditions [3,4,5]. However, it has remained a puzzle as to why some individuals with a high-risk factor exposure live long and relatively healthy lives while others with comparably low or similar exposures experience greater morbidity and premature death. A plausible explanation is that different individuals are more or less vulnerable to certain risk factors and the underlying pathological processes of these risk factors. As longevity exhibits high heritability [6], investigating how genetic variability may contribute to promoting health and reducing the risk of diseases may improve our understanding of underlying biological mechanisms. However, to date, only a few genes and genetic loci have been identified for this trait [7], and therefore the study of genes contributing to longevity and vulnerability to disease is a developing science.

Major physiopathological mechanisms thought to underlie the deleterious effects of risk factors for cognitive decline, neurodegeneration, and ageing more generally, are oxidative stress (OS) and chronic inflammation [8,9,10]. OS and inflammation can contribute in various ways to disease and ageing processes, of which DNA damage is a major demonstrated pathway [11,12,13]. DNA repair is a primary hallmark of ageing [14]; therefore, a possible explanation for variability in vulnerability to risk factor exposure is that the efficiency of DNA repair mechanisms varies enough between individuals to have a substantive impact on their health trajectories and vulnerability to neurodegenerative diseases [15,16].

The type of DNA damage caused by OS includes oxidized DNA nucleotides, single-strand breaks (SSBs), double-strand breaks (DSBs), and telomere shortening [17,18]. To prevent catastrophic outcomes linked to such damage (e.g., cellular death, cancers), humans possess a number of DNA repair mechanisms [19,20]. They include homologous recombination, non-homologous end joining, base excision repair (BER), nucleotide excision repair (NER), and mismatch repair (MMR), as well as DNA damage detection mechanisms such as Poly ADP Ribose Polymerase (PARP). There is variation (single nucleotide polymorphism; SNP) in the genes that code for these repair mechanisms between individuals; however, SNPs have mostly been investigated in relation to their role in cancers [21]. Based on the limited existing evidence, BER and NER are the repair mechanisms whose genetic variability appears to be most relevant to the brain and cognitive health [22,23]. This is because the brain is the organ with the highest metabolic rate and, as such, produces a high amount of reactive oxygen species (ROS) [24,25]. As in other parts of the body, a large proportion of ROS are buffered by antioxidants. However, the remainder may contribute to proportionally higher OS levels and related DNA damage. Indeed, elevated levels of DNA damage in the CNS of animal models have been reported [15]. Moreover, in post-mitotic cells, such as neurons, DNA damage is not mitigated by the more robust DNA repair mechanisms involved in cell replication but by the less reliable DNA repair mechanisms involved in transcription, to which BER and NER also contribute [26]. Thus, it would be expected that variability in efficiency, particularly for BER, given it is the major pathway to repair oxidative DNA damage [15,26,27], leads to more or less damage accumulating in the brain. Such differences in damage accumulation and increased vulnerability to oxidative agents have been demonstrated in cell cultures [28], and variability in BER genetic variants was found to be associated with increased DNA damage in Alzheimer’s disease patients compared to the controls [29]. However, we currently do not know how much such variability contributes to age-related differences in cognitive function or brain health.

The objective of this study is to determine whether genetic variability (SNPs) in the DNA repair mechanism contributing most to resolving OS-related damage (BER), is associated with cerebral and cognitive health.

## 2. Materials and Methods

### 2.1. Study Population

All participants taking part in the UK Biobank study (UKB) were considered for inclusion. The UKB has been described elsewhere in detail [21,22], but briefly, the UKB is a prospective cohort study of 502,655 participants aged 37 to 73 years at the baseline who were assessed across 22 assessment centers around the UK between 2006 and 2019.

Participants for whom genomic data (*n* = 488,377), at least one cognitive measure (*n* = 496,542; numbers varied for different measures), and essential covariates (age, sex, education; *n* = 492,236) were available were included for analysis providing a total sample of 488,013 participants (Appendix A). In addition, a subset of participants who undertook a brain scan (*n* = 39,060) contributed to neuroimaging analyses. This study follows the Strengthening the Reporting of Observational Studies in Epidemiology (STROBE) guidelines [23].

### 2.2. DNA Repair Single Nucleotide Polymorphism

BER SNPs were selected in two steps. First, all SNPs of BER genes with published significant associations with cognitive or neurological outcomes documented in SNPedia were identified (*OGG1*, *NEIL1-2-3*, *MUTYH*, *NTHL1*). Second, those identified SNPs which were also available in the UK Biobank genome-wide data archive were considered for analysis (*n* = 17; *OGG1*: rs1052133, rs104893751 *NEIL1*: rs7402844, rs5745906 *NEIL2*: rs6601606 *NEIL3*: rs10013040, rs13112390, rs13112358, rs1395479 *MUTYH*: rs34612342, rs200165598, rs77542170, rs200844166, rs200495564, rs121908381 *NTHL1*: rs150766139, rs2516739). Of those, four were excluded due to low prevalence (*n* < 50; rs77542170, rs200844166, rs200495564, rs121908381), leaving 13 SNPs for analysis (Appendix A). The SNPs selected for analysis and the function of the genes they belong to are presented in Table 1.

### 2.3. Cognitive Measures

Fluid intelligence (FIQ; UKB field 20191): Participants responded to as many as 14 multiple-choice questions, testing as many numerical, logic, arithmetic, and syntactic skills as possible in two minutes. Scores were computed as the sum of correct answers (range 0–14).

Symbol-digit matching test (SDMT; UKB field 20159): Participants were presented with a series of grids in which symbols had to be matched to numbers according to a key presented on the screen. Scores were the number of correct answers (range 0–103).

Pair matching test (MATCH; UKB field 20023): Participants were presented with 6 or 12 cards on a screen with their faces concealed. Each pair was revealed one by one for 5 s and then turned over again. Scores were the time (milliseconds) taken to correctly identify matches (range 63–2000).

Trail-Making 1 and 2 (TRAIL1 and TRAIL2; UKB fields 20156 and 20157): The participant was presented with sets of digits (TRAIL1) or digits and letters (TRAIL2) in circles scattered around the screen. In TRAIL1, they were asked to sequentially click on digits. In TRAIL2, they were asked to sequentially click on digits and letters, alternating between each (i.e., A—1, B—2, etc.). Scores were the time (seconds) to complete the numeric or α-numeric paths (range TRAIL1: 13-734; TRAIL2: 20-746).

### 2.4. Brain Measures

#### 2.4.1. Image Acquisition

Participants underwent an MRI scan during the second (2014+) visit at one of three imaging centers using the same scanner (3T Siemens Skyra, running VD13A SP4 using a 32-channel head coil; Siemens Healthcare, Erlangen, Germany). Detailed UKB imaging protocols are provided online [21]. Briefly, all participants were imaged with a T1-weighted 3D magnetization-prepared rapid acquisition gradient echo sequence over a five-minute duration in the sagittal orientation (resolution = 1 × 1 × 1 mm; field of view = 208 × 256 × 256 matrix).

#### 2.4.2. Segmentation and Image Analysis

FreeSurfer was used to segment and analyze neuroimaging data (version 6.0.5) [27]. The FreeSurfer pipeline has been extensively described elsewhere [28]. In summary, it involves motion correction, transformation to the Talairach image space, inhomogeneity correction, non-brain tissue removal using a hybrid watershed, volumetric segmentation [29,30], and cortical surface reconstruction and parcellation [21,31]. The regions of interest (ROIs) selected a priori to the analysis were total grey matter (GM), total white matter (WM), the left (LHC) and right hippocampus (RHC), and white matter lesions (WMLs) because these areas are either highly vulnerable to risk factors associated with accelerated ageing or because they are likely to best capture diffuse effects across the brain [30,31,32].

### 2.5. Socio-Demographic and Health Measures

Age, sex, education, smoking, alcohol intake, type 2 diabetes, and heart problems were obtained via self-report. Education was recoded into primary, secondary, professional certificate/diploma, tertiary, or unknown. Body mass index (BMI) was computed with the following formula: weight (kg)/height (m^2^).

### 2.6. Statistical Analysis

Statistical analyses were computed using the R statistical package (version 4.2.3) in RStudio (version 2023.03.0). Associations between selected SNPs and cognitive outcomes were tested through linear regression analyses controlling for age, sex, and education. Univariate models testing each SNP separately were first conducted to determine whether individual SNPs were significantly associated with outcome measures. Next, multivariate models, including all SNPs with significant associations detected in univariate analyses with one or more cognitive outcomes, were conducted. The models were progressively reduced (using stepwise deletion) to only include those SNPs that remained significant. This was performed to identify the most influential SNPs. To further determine whether pairs of SNPs interacted together, SNPs with significant associations were combined pairwise by creating categorical variables reflecting all allele permutations, and additional regression models were computed only for the cognitive measure with the largest sample size (MATCH). Next, significant SNPs were classified as protective if they were associated with better cognitive function or as a risk if they were associated with poorer cognitive function. Based on this classification, the following three indexes were computed: a protective index (protect) computed as the count of protective SNPs an individual has, a harm index (harm) computed as the count of risk SNPs, and an overall risk index computed by subtracting the protective index from the harm index (risk) and shifting its range so the lowest risk was 0. Associations between the three indexes, cognitive outcomes, and brain measures were tested with similar linear models to those used in univariate analyses (brain analyses were also controlled for ICV) while also including interaction terms for age categories (middle-age: <60 years; older-age: ≥60 years) and sex. Finally, to further determine whether influential SNPs were clustered in groups of individuals and to assess whether such clusters were differentially associated with cognitive and brain health, a latent class analysis (LCA) was conducted with the poLCA R package (version 1.6.0.1) [33]. The LCA was restricted to the 12 SNPs for which one or more significant associations were detected, and SNP variables were transformed into binary factors (none vs. one/two alleles); pilot analyses showed that this produced essentially the same results with a better fit. Associations between identified classes and the cognitive and brain measures were then tested with the same type of regression models as those used in multivariate analyses. α was set at *p* < 0.05 and corrected for multiple comparisons (Bonferroni).

## 3. Results

Participants’ demographic characteristics are presented in Table 2, with ethnic origin information in Appendix A and the bivariate Pearson correlation between outcome variables and covariates in Appendix A.

### 3.1. SNPs—Cognition Associations

Univariate associations between SNPs and cognitive measures presented in Appendix A demonstrate that most SNPs have significant associations with one or more outcomes. Five SNPs (*OGG1*: rs104893751; *NEIL1*: rs7402844; *NEIL3*: rs13112390, rs13112358, rs1395479) were associated with better function and were labeled “protective”. Three SNPs (*OGG1*: rs1052133; *NEIL2*: rs6601606; *NTHL1*: rs2516739) were associated with lower function and were labeled “harmful”. Two SNPs (*NTHL1*: rs150766139; *MUTYH*: rs200165598) did not contribute to any significant associations.

In the multivariate analysis, the reduced model (Table 3) shows that after the stepwise deletion of non-significant predictors, eight SNPs (protective: *NEIL1*: rs7402844, *NEIL3*: rs13112358, *NEIL3*: rs1395479; harmful: *OGG1*: rs1052133, *NEIL2*: rs6601606, *NTHL1*: rs2516739, *MUTYH*: rs200165598) were found to be independently associated with cognitive function. For most SNPs, the presence of the variant in the two alleles (homozygosity) produced stronger effects ranging in magnitude between 27.5% and 355.7% above those observed when the variant was present in a single allele (heterozygosity). The strongest and numerous associations were detected for the MATCH measure, which had the largest sample size, and for the FIQ measure.

Associations between the pairwise combination of significant SNPs identified in the multivariate model and cognitive measures are presented in Appendix A. The SNP combination resulted mostly in a subtractive effect, while some combinations resulted in apparently additive and synergetic effects. The strongest subtractive effects were detected for *NEIL1*: rs7402844 and *OGG1*: 1052133, *OGG1*: rs1052133 and *NEIL3*: rs13112358, *OGG1*: rs1052133 and rs13954791, *NEIL1*: rs7402844 and *NEIL2*: rs6601606, *NEIL1*: rs7402844 and *NTHL1*: rs2516739, *NTHL1*: rs2516739 and *NEIL3*: rs13112358; the strongest additive effects were detected for *NEIL1*: rs7402844 and *NEIL3*: rs13112358, *NEIL1*: rs7402844 and *NEIL3*: rs1395479; the strongest synergetic effects were detected for *NEIL2*: rs6601606 and *OGG1*: rs1052133, *NEIL2*: rs6601606 and *NTHL1*: rs2516739.

### 3.2. Protect, Harm, and Risk Indexes

The characteristics of the protect, harm, and risk indexes are presented in Appendix A. Associations between these indexes, as well as their interactions with age and sex, and the MATCH cognitive measure (selected due to a larger sample size), are presented in Table 4. All indexes were significantly associated with cognitive function in the predicted direction. In addition, significant two-way interactions with age were detected for all indexes such that higher risk was associated with lower performance (Figure 1). No significant interactions were detected for sex.

### 3.3. SNPs—Brain Associations

The reduced multivariate model testing associations between the twelve significant SNPs identified above and brain volumes (Table 5) shows that after the stepwise removal of non-significant predictors, 2 SNPs (*NEIL3*: rs13112390, rs13112358; 1 protective and 1 harmful) were significantly associated with the total GM only, with the presence of the variant in the two alleles producing stronger associations.

In addition, significant associations were detected between the risk index and most brain volumes (Appendix A), revealing three-way interactions between risk, age, and sex. These interactions indicate that while the risk is not substantially associated with brain volume in middle-aged participants, it is positively associated with LHC, RHC, GM, and WMH (in the opposite direction) volumes in females and negatively in males (see also Figure 2).

### 3.4. Latent Class Analyses

A two-class model identifying one larger class (95.4%) and a smaller class (4.6%) was selected as it best fitted the data and because the three-class model did not converge (see Appendix A). Class membership was primarily based on three SNPs (*NEIL3*: rs13112358, rs13112390, rs10013040), which were present in class 2 but either not or to a much lesser extent in class 1. Other SNPs contributed little to defining class membership. Associations between classes and cognitive outcomes are presented in Appendix A and indicate that the second class was associated with an overall worse cognitive function (lower FIQ, higher MATCH scores). In addition, strong three-way interactions with age and sex were detected, indicating that a stronger effect is present in middle-aged women (Appendix A), following a similar pattern to that observed above in relation to the risk measure. No main effect was present for brain measures, but three-way interactions with age and sex were also detected, which are suggestive of effects, trending in the opposite direction for LHC, RHC, and GM in men and women, which also appear to be modulated by age (Appendix A).

## 4. Discussion

The findings from this study provide compelling evidence that variability in genes involved in DNA repair is associated with cognitive function and, to a lesser extent, with brain structure. Indeed, whether investigating associations between the genetic variability of selected genes and cognitive function in univariate or multivariate analyses or when considering their combined effect in a risk index or through latent class analysis, consistent results emerged. Importantly, this was in spite of a very stringent statistical correction (Bonferroni) being applied.

In univariate analyses, which tested each variant individually, all but two SNPs were significantly associated with cognitive performance, with five suggestive of a protective effect and three of a harmful effect. By contrast, in multi-variate analyses, which tested all variants found to have significant associations in univariate analyses, significant associations after model reduction were only found for eight SNPs, with three suggestive of a protective effect and four of a harmful effect. Where sufficient variance was available to detect an allele dose effect, the presence of two alleles was generally associated with a greater effect ranging in magnitude between 27.5% and 355.7% in multivariate analyses. These effects were predominantly observed with the matching task and fluid intelligence measures. This was not unexpected for the matching task since it has the greatest range and was available for the largest sample size. However, the range of the fluid intelligence measure was much narrower, and the strong associations detected suggest that the variability in DNA repair mechanisms might have an influence on a broad range of cognitive processes.

The strongest and most consistent associations were detected with SNPs of the *NEIL1* (rs7402844), *NEIL2* (rs6601606), and *NTHL1* (rs2516739) genes, which also displayed an allele dose effect. This is interesting because a role for BER in cognitive function has been reported both in rodents and humans. In mice, the loss of *NEIL1* causes deficits in olfactory function and short-term spatial memory retention. By contrast, *Neil1^−/−^Neil2^−/−^*-deficient animals display hyperactivity, reduced anxiety, and improved learning [34,35,36]. Moreover, *NEIL1* deficiency impairs the survival of newly generated hippocampal neurons and memory performance in young adult male mice [37]. In humans, *NEIL1* SNPs have been associated with late-onset Alzheimer’s disease [38], and *NEIL1* gene expression is down-regulated in the lymphocytes of AD patients [39].

Additional analyses aimed at determining whether there were synergetic effects between pairs of SNPs found some support for this hypothesis for *NEIL2*: rs6601606 and *OGG1*: rs1052133, and *NEIL2*: rs6601606 and *NTHL1*: rs2516739. Other SNPs tended to have an additive effect if they were both labeled as protective or both labeled as harmful and a subtractive effect otherwise. Such incremental effects were further demonstrated when genes implicated in significant associations were combined in a risk index. Indeed, having a larger risk index was also associated with lower cognitive performance, particularly at older ages.

Given the high statistical power afforded by the very large sample size used in this study, it is important to critically consider the practical importance of these findings, as it is widely recognized that statistical significance is not equivalent to functional relevance in a broader context. While effect sizes varied substantially across the genes implicated in significant associations, compared to individuals who did not carry particular SNPs, those who carried one allele had absolute differences in cognitive function ranging from <1% to 5% both in univariate and multivariate analyses. Similarly, the synergetic effects detected between pairs of genes were associated with a difference in function ranging from ~2% to 6% in those who had one allele of each SNP compared to none. These are very substantial differences, particularly when considering the measurement noise inherent to such epidemiological studies.

To further establish whether influential SNPs cluster in the same individuals and, thus, potentially lead to even more potent effects, we conducted a latent class analysis on 12 SNPs for which significant differences were detected in prior analyses. The LCA yielded two classes, which were almost exclusively defined by the presence of variants in three SNPs of the *NEIL3* gene in the second, smaller class. The association between class and cognition varied by age and sex, such that belonging to the second class was associated with the greatest difference in fluid intelligence (−4.1%) in middle-aged women. In contrast, associations in men and in older individuals, while following the same direction, were weaker. The origin of this age and sex difference is unclear. A possible explanation is that the three variants, all belonging to NEIL3, which discriminate between class 1 and class 2, somehow interact with female sex hormones. Intriguingly, a recent study [40] in an Alzheimer mouse model deficient for NEIL3 showed an age-dependent decrease in amyloid β plaque deposition in females, which was not observed in males and which was not attributable to increased DNA damage as similar levels were observed in the two sexes. This study did not investigate the NEIL3 variants but may suggest that the efficiency in this gene somehow interacts with female sex hormones to influence cognitive function.

Importantly, the findings discussed above were not only observed in relation to cognitive function but also, to a lesser extent, with brain volumes. A three-way interaction between age, sex, and the risk index indicated that while increasing risk has minimal associations with hippocampal and grey matter volume in middle-aged participants, it contributes to stronger associations and different directions in older men and women. It is not clear why a higher risk is associated with larger volumes in older women and lower volumes in older men. Increases in volumes at older ages have been reported in previous research, including in post-menopausal women [31], and it may reflect the increased neuroinflammation that precedes neurodegeneration [41]. This explanation is also consistent with our findings, indicating that greater risk at all ages and in both sexes is associated with lower cognitive performance. Interestingly, recent research has demonstrated that the expression of a number of BER glycosylases (NEIL1, NEIL2, OGG1, and NTH1) varies in the brain by age and region [27]. Moreover, ROS production also appears to differ across brain regions [24]. Thus, age- and regions-specific variation in OS and DNA repair may explain varying patterns of associations with brain volumes and cognitive functions with increasing age.

### Limitations

This study has several strengths as well as limitations. The very large sample size ensured that sufficient statistical power was available to detect small effects and made it possible to test complex interactions. However, it should be noted that sample sizes were substantially reduced for brain analyses, which could explain the presence of weaker associations with brain volumes. Another strength is that SNPs were selected a priori based on past research, implicating them in ageing-related processes, and analyses were robustly controlled for multiple comparisons, thus substantially reducing the risk of a type 1 error. However, only a relatively small number of SNPs, which were easily available in the UKB data library, could be investigated. Therefore, well-resourced future investigations should aim to extract relevant SNPs directly from whole-genome data to extend the present findings. In addition, while the UKB is broadly representative of the UK population, it may not be reflective of other world regions and ethnicities, and, therefore, the generalization of the present findings should be considered with caution. As always, correlational research such as this one cannot infer causation.

## 5. Conclusions

The present findings provide convincing evidence that the variability in genes implicated in DNA repair, particularly *NEIL1*, *NEIL2*, *NTHL1*, and *OGG1*, are associated with cognitive function and brain structure in ageing. These effects are substantial, as variants were found to be associated with up to 6% differences in the performance of a pair-matching task. This suggests that the variants investigated moderate the efficiency of DNA repair mechanisms coded by BER genes. These differences in efficiency may lead to the differential accumulation of DNA damage in brain cells and cause mutations in genes important for brain function, which ultimately could impact individuals’ cognitive function and the underlying brain structure to varying degrees. However, the exact mechanisms involved are unknown and warrant further investigation. These findings are particularly important because the variants investigated are not rare, with the most influential ones being present in between 3% and 40% of the population studied. There is, therefore, a pressing need for large-scale, holistic investigations of the various roles in DNA repair and other related mechanisms that contribute to heterogeneous ageing trajectories, as well as a functional analysis of the variants in DNA repair genes that affect cognition. Future research should also aim to clarify how differences in the efficiency of DNA repair interact with environmental exposures, with a particular focus on modifiable lifestyle and health risk factors, as a better understanding of their independent and synergetic effects is likely to provide important insights that may inform the prevention and identification of individuals at risk and the development of new pharmaceuticals.

## Figures and Tables

**Figure 1 genes-15-00153-f001:**
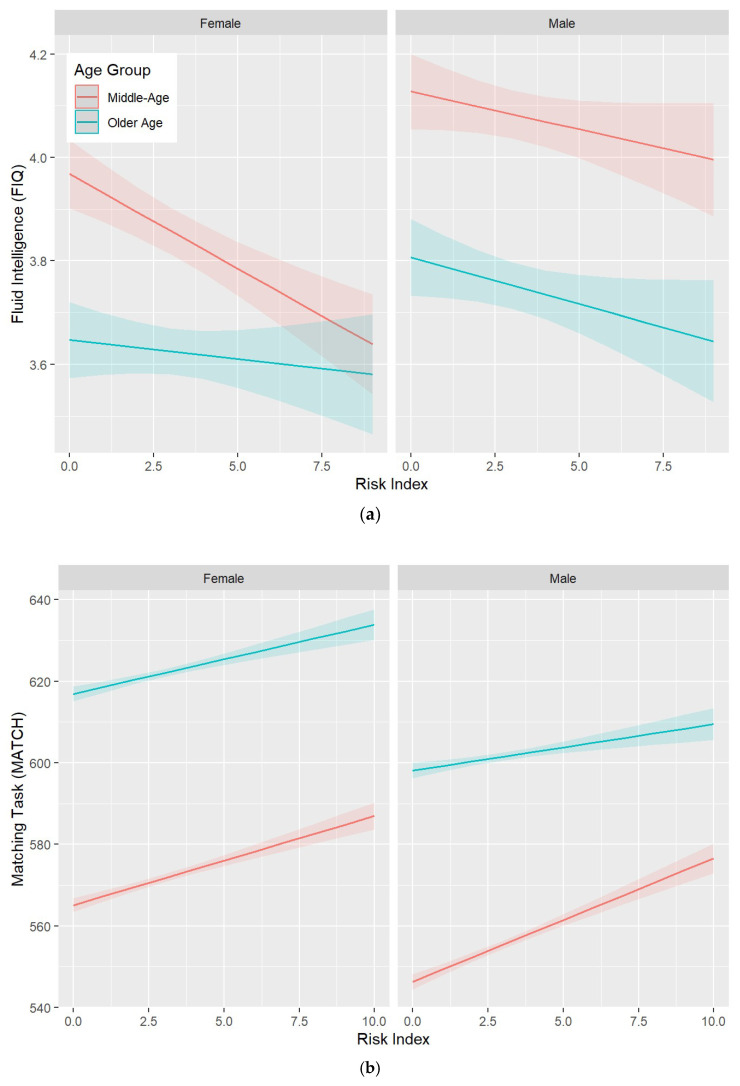
Three-way interactions between the risk index, age category (middle-age > 60 years vs. older-age ≥ 60 years), sex, and (**a**) the FIQ and (**b**) MATCH cognitive measures. For FIQ, the significant interaction indicates that performance decreases more steeply with an increasing risk index for middle-aged women than for older women or men. For MATCH, the significant interaction indicates that response time increases more steeply for older men than middle-aged men. Shaded areas indicate 95% confidence intervals.

**Figure 2 genes-15-00153-f002:**
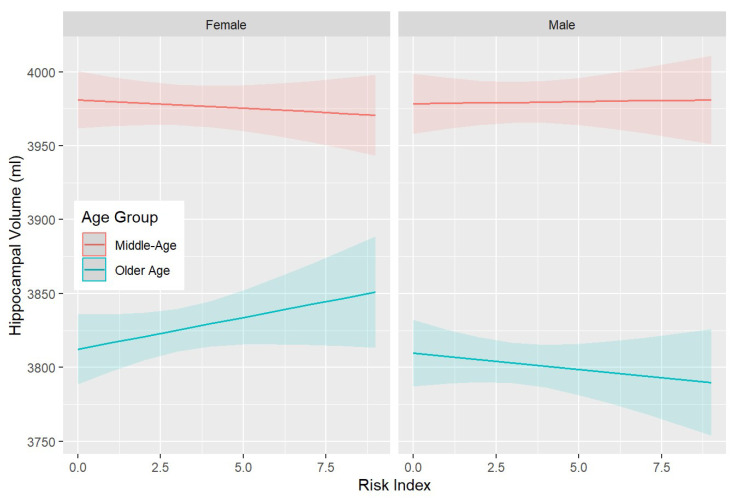
Three-way interactions between the risk index, age category (middle-age > 60 years vs. older-age ≥ 60 years), and sex in predicting the left hippocampal volume (LHC). The significant interaction indicates that where hippocampal volume increases with an increasing risk index in older women, the opposite association is observed in older men. Shaded areas indicate 95% confidence intervals.

**Table 1 genes-15-00153-t001:** Summary of the base excision repair genes investigated, their functional role, and the specific SNPs considered in the present study.

Genes	Function in Base Excision Repair	SNPs
* **OGG1** *	Detection and excision of pyrimidines in double-stranded DNA, including the most frequently occurring oxidized DNA lesion, 8-oxoguanine.	rs1052133 rs104893751
* **NEIL1** *	Detection and excision of pyrimidines, 8-oxoguanine, and formamidopyrimidine from both single and double-stranded DNA, preferably in bubble-structured DNA, as well as in close proximity to another DNA lesion. Contributes particularly to transcription-associated DNA repair.	rs7402844 rs5745906
* **NEIL2** *	Similar to *NEIL1*, however, its expression is cell-cycle independent with a particular affinity for cytosine-derived lesions such as 5-hydroxyuracil.	rs6601606
* **NEIL3** *	Similar to *NEIL1* and *NEIL2*, but mostly expressed in development, e.g., in brain regions rich in progenitor cells (subventricular zone, hippocampus, cerebellum), decreasing with age.	rs10013040 rs13112390 rs13112358 rs1395479
* **MUTYH** *	Provides protection against the mispairing of adenosine with 8-oxoguanine by removing the adenosine base and prevents the accumulation of 8-oxoguanine lesions. *MUTYH* can also remove other sources of lesions, such as oxidized adenines.	rs34612342 rs200165598
* **NHTHL1** *	Detection and excision of pyrimidines and purines. Specific protective action against telomeric lesions.	rs150766139 rs2516739

Bold: OGG1: 8-Oxoguanine glycosylase; NEIL1/2/3: NEI endonuclease VII like 1/2/3; MUTYH: mutY DNA glycosylase (human); NHTHL1: Nth Like DNA Glycosylase 1.

**Table 2 genes-15-00153-t002:** Participants’ characteristics for female, male, and all participants. Sample sizes are presented in square brackets. Standard deviation (SD) for continuous measures and percentages for count measures are presented in round brackets. BMI: body mass index.

	Female [*n* = 264,576]	Male [*n* = 223,437]	Total [*n* = 488,013]
**Age (years)**			
Mean (SD)	56.36 (8.00)	56.75 (8.20)	56.54 (8.09)
**Education (highest qualification)**			
Primary	44,727 (16.9%)	38,546 (17.3%)	83,273 (17.1%)
Secondary	107,069 (40.5%)	76,355 (34.2%)	183,424 (37.6%)
Prof certificate/diploma	27,067 (10.2%)	30,204 (13.5%)	57,271 (11.7%)
Tertiary	82,457 (31.2%)	75,318 (33.7%)	157,775 (32.3%)
Unknown	3256 (1.2%)	3014 (1.3%)	6270 (1.3%)
**Smoke (ever)**			
No	117,855 (44.8%)	77,159 (34.7%)	195,014 (40.2%)
Yes	145,436 (55.2%)	145,119 (65.3%)	290,555 (59.8%)
**Alcohol**			
Current	238,953 (90.3%)	208,754 (93.4%)	447,707 (91.7%)
Never	15,359 (5.8%)	6193 (2.8%)	21,552 (4.4%)
Past	9631 (3.6%)	7893 (3.5%)	17,524 (3.6%)
Unknown	633 (0.2%)	597 (0.3%)	1230 (0.3%)
**BMI (kg/m^2^)**			
Mean (SD)	27.07 (5.18)	27.83 (4.24)	27.42 (4.79)
Range	12.12–74.68	12.81–68.41	12.12–74.68
**Diabetes**			
No	253,463 (95.8%)	206,618 (92.5%)	460,081 (94.3%)
Yes	10,177 (3.8%)	15,718 (7.0%)	25,895 (5.3%)
Unknown	936 (0.4%)	1101 (0.5%)	2037 (0.4%)

**Table 3 genes-15-00153-t003:** Associations between SNPs and cognitive outcomes assessed in a single multivariate model, including only those SNPs with significant univariate associations detected in analyses testing the same effects in individual SNPs one at a time (see Appendix A). Digits (1 or 2) in brackets indicate the effect of the heterozygote and homozygote carriers (vs. non-carriers) of each SNP except for rs200165598, for which only heterozygote carriers were present in the sample. The statistics reported include unstandardized β estimates and *p* values.

	Cognitive Measures				
SNP Variants (1/2 Alleles vs. None)	FIQ	SDMT	MATCH	TRAIL1	TRAIL2
rs1052133(1)			0.669		
			*p* = 0.052		
rs1052133(2)			2.322 **		
			*p* = 0.002		
rs7402844(1)	0.080 **		−8.690 ***		−0.661 *
	*p* = 0.001		*p* < 0.00001		*p* = 0.034
rs7402844(2)	0.102 ***		−10.890 ***		−0.921 **
	*p* = 0.00001		*p* = 0.000		*p* = 0.003
rs6601606(1)	−0.187 ***		6.889 ***		
	*p* < 0.00001		*p* < 0.00001		
rs6601606(2)	−1.559 ***		24.501 **		
	*p* < 0.00001		*p* = 0.0004		
rs13112358(1)	0.081 **		−2.537 **		
	*p* = 0.001		*p* = 0.0004		
rs13112358(2)	0.094 **		−4.148 ***		
	*p* = 0.0001		*p* < 0.00001		
rs2516739(1)	−0.003		0.899 *		
	*p* = 0.811		*p* = 0.010		
rs2516739(2)	−0.079 **		5.669 ***		
	*p* = 0.003		*p* < 0.00001		
rs1395479(1)		0.115 **	−2.809 ***		
		*p* = 0.0001	*p* < 0.00001		
rs1395479(2)		0.077	−3.556 ***		
		*p* = 0.163	*p* < 0.00001		
rs200165598(1)			18.339 **		
			*p* = 0.003		
Constant	3.536 ***	16.337 ***	614.845 ***	47.156 ***	87.129 ***
	*p* < 0.00001	*p* < 0.00001	*p* < 0.00001	*p* < 0.00001	*p* < 0.00001
Observations	120,453	115,893	478,185	101,909	101,788
Log Likelihood	−248,761	−342,715	−2,935,306	−415,735	−466,588
Akaike Inf. Crit.	497,552	685,449	5,870,652	831,487	933,196

Note: * *p *< 0.05; ** *p *< 0.0035; *** *p* < 0.00001.

**Table 4 genes-15-00153-t004:** Associations between the protect, harm, and risk indexes and performance on the MATCH cognitive measure. The protect index is the total number of SNPs with a protective effect carried by each individual, while the harm index is the total number of SNPs with a harmful effect, and the risk index is the difference between the protect and harm indexes. The statistics reported include unstandardized β estimates and *p* values.

Dependent Variable		MATCH	
	Protect	Harm	Risk
Age (years)	3.136 ***	3.668 ***	3.978 ***
	*p* < 0.00001	*p* < 0.00001	*p* < 0.00001
Sex (Male)	−16.503 ***	−18.113 ***	−17.666 ***
	*p* < 0.00001	*p* < 0.00001	*p* < 0.00001
Index	−2.050 ***	1.315 ***	1.684 ***
	*p* < 0.00001	*p* = 0.00005	*p* < 0.00001
Age × Index	0.135 ***	−0.75 ***	−0.109 ***
	*p* < 0.00001	*p* = 0.008	*p* < 0.00001
Sex × Index	−0.387	0.162	0.364
	*p* = 0.295	*p* = 0.722	*p* = 0.204
Constant	609.515 ***	600.920 ***	596.709 ***
	*p* < 0.00001	*p* < 0.00001	*p* < 0.00001
Observations	476,240	480,300	473,791
Log Likelihood	−2,923,622	−2,948,533	−2,908,645
Akaike Inf. Crit.	5,847,265	5,897,086	5,817,310

Note: * *p *< 0.1; ** *p* < 0.05; *** *p* < 0.01.

**Table 5 genes-15-00153-t005:** Associations between SNPs and brain volumes assessed in a single multivariate model, including only those SNPs with significant univariate associations and cognitive measures (see Appendix A). Digits (1 or 2) in brackets indicate the effect of the heterozygote and homozygote carriers (vs. non-carriers) of each SNP except for rs34612342 and rs200495564, for which only heterozygote carriers were present in the MRI subsample. The statistics reported include unstandardized β estimates and *p* values. Left (LHC) and right (RHC) hippocampus; grey (GM) and white (WM) matter; white matter hyperintensities (WMHs).

	Brain Volumes					
SNP Variants (1/2 Alleles vs. None)	LHC	RHC	GM	WM	WMH	
Age (years)	−12.031 ***	−12.582 ***	−753.194 ***	−138.614 ***	135.550 ***	
	*p* < 0.00001	*p* < 0.00001	*p* < 0.00001	*p* < 0.00001	*p* < 0.00001	
Sex (Male)	3.885	−14.718 ***	3426.919 ***	−4741.967 ***	−163.435 ***	
	*p* = 0.321	*p* = 0.0004	*p* < 0.00001	*p* < 0.00001	*p* < 0.00001	
rs34612342(1)	43.861 *	34.667		2674.997 *		
	*p* = 0.052	*p* = 0.145		*p* = 0.071		
rs200495564(1)	−472.157	−137.377				
	*p* = 0.126	*p* = 0.674				
rs13112390(1)			−2476.289 ***			
			*p* = 0.009			
rs13112390(2)			−3008.684 ***			
			*p* = 0.004			
rs13112358(1)			1914.805 **			
			*p* = 0.022			
rs13112358(2)			2233.274 **			
			*p* = 0.019			
rs6601606(1)					−40.886	
					*p* = 0.598	
rs6601606(2)					1219.892 *	
					*p* = 0.059	
Constant	1473.881 ***	1503.877 ***	104,839.200 ***	−81,903.510 ***	−2138.101 ***	
	*p* < 0.00001	*p* < 0.00001	*p* < 0.00001	*p* < 0.00001	*p* < 0.00001	
Observations	38,991	38,991	38,781	39,025	39,044	
Log Likelihood	−278,780	−280,983	−438,138	−442,333.300	−362,143	
Akaike Inf. Crit.	557,572	561,979	876,293	884,676.600	724,299	

Note: * *p *< 0.1; ** *p* < 0.05; *** *p* < 0.01.

## Data Availability

The data that support the findings of this study are available from the UK Biobank; however, restrictions apply to the availability of these data, which were used under license for the current study, and so are not publicly available. Data are, however, available from the authors upon reasonable request and with the permission of the UK Biobank.

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
