# Peer review of "Cognitive Function Is Associated with the Genetically Determined Efficiency of DNA Repair Mechanisms"

_genes, 2024, doi:10.3390/genes15020153_

Round 1
Reviewer 1 Report
Comments and Suggestions for Authors
In their paper, Cherbuin et al. investigated the association between 12 SNP in genes coding for BER and higher cognition processes. This is a serious and interesting study. The paper is well written and the methodology is sounds. I have only minor remarques. The authors must describe better the legend of the main figures. Most of the legend are at the moment NOT self supporting. The data are very dense, and it will ease the reading if the authors help the reader to understand better the different results. For example, in Table 1 what are the numbers in brackets? If this is the different alleles, this should be indicated, and make a remark on the last SNP for which a single allele is analysed (because of the coverage?). What is the statistical test and what is the criteria for significancy. This must be indicated in the legend for the main figures and table; the authors can not only refeer to supplementary material. Such careful description of the legends, must be done for all figures and tables and especially for the main text.
Below are some other comments:
Line 74 : “DNA repair is a primary hallmark of ageing”. The cited review does not specifically describe such process. Perhaps it is better to say: “Decreased genome stability is a primary hallmark of ageing”. Or cite another reference to support the statement “Reduced DNA repair is a ….”
Line 81: change to ((e.g. genomic instability, cellular death, cancers)
Line 156: “Age, sex education, smoking, type 2 diabetes”. I wonder why sex education is a factor here? Or perhaps a comma is missing, which make a very funny interpretation. I would have asked for alcohol consumption which is not described here. Please provide information if alcohol consumption was checked and otherwise indicate this point in the Material and Methods.
The legend of Figure 2. The legend if this figure is very minimalistic. Please describe better this figure. The significancy of each association should be indicated on the graph.
Line 292: If I am not mistaking, the allele dose-effects if several SNP in BER is not presented in the results section. Please provide some information on this point in Results, otherwise the discussion is not supported by results.
Author Response
R1
In their paper, Cherbuin et al. investigated the association between 12 SNP in genes coding for BER and higher cognition processes. This is a serious and interesting study. The paper is well written and the methodology is sounds. I have only minor remarques. The authors must describe better the legend of the main figures. Most of the legend are at the moment NOT self supporting. The data are very dense, and it will ease the reading if the authors help the reader to understand better the different results. For example, in Table 1 what are the numbers in brackets? If this is the different alleles, this should be indicated, and make a remark on the last SNP for which a single allele is analysed (because of the coverage?). What is the statistical test and what is the criteria for significancy. This must be indicated in the legend for the main figures and table; the authors can not only refeer to supplementary material. Such careful description of the legends, must be done for all figures and tables and especially for the main text.
Response:
Thank you for your careful review and helpful suggestions. We have addressed all of them and paid particular attention to providing clearer and more detailed legends for all tables and figures in the main manuscript as well as in the supplementary material.
Below are some other comments:
Point 1:
Line 74 : “DNA repair is a primary hallmark of ageing”. The cited review does not specifically describe such process. Perhaps it is better to say: “Decreased genome stability is a primary hallmark of ageing”. Or cite another reference to support the statement “Reduced DNA repair is a ….”
Response:
As suggested, we now refer to “Decreased genome stability” (Line 74).
Point 2:
Line 81: change to ((e.g. genomic instability, cellular death, cancers)
Response:
Implemented as suggested (Line 81).
Point 3:
Line 156: “Age, sex education, smoking, type 2 diabetes”. I wonder why sex education is a factor here? Or perhaps a comma is missing, which make a very funny interpretation. I would have asked for alcohol consumption which is not described here. Please provide information if alcohol consumption was checked and otherwise indicate this point in the Material and Methods.
Response:
Thank you for alerting us to this grammatical error; the missing comma has been added. Alcohol intake has been added to the method and to Table 1.
Point 4:
The legend of Figure 2. The legend if this figure is very minimalistic. Please describe better this figure. The significancy of each association should be indicated on the graph.
Response:
As requested, we have improved the legend of Figure 2 (as well as those of all other figures) by describing the key drivers of the significant interaction and by reporting that the shaded areas represent the 95% confidence intervals. We have not added statistics to the figure because they are already presented in the corresponding Tables and because the 95%CI already clearly shows which conditions are significantly different from one another.
Point 5:
Line 292: If I am not mistaking, the allele dose-effects if several SNP in BER is not presented in the results section. Please provide some information on this point in Results, otherwise the discussion is not supported by results.
Response:
The results do refer to the dose-effects both in text “For most SNPs the presence of the variant in the two alleles produced stronger associations.” (line 210) and in Table 2, which this statement refers to, where effect sizes are reported.
This reviewer likely refers to the fact that in the discussion we comment on these results as “Where sufficient variance was available to detect an allele dose-effect, having two alleles was generally associated with a greater effect ranging in magnitude between 27.5% and 355.7% in multi-variate analyses”. The percentage is a reformulation of the effect sizes presented in Table 2, but we concede we did not report this in text in the results section. We have now amended the manuscript to clarify this point as follows:
“For most SNPs the presence of the variant in the two alleles (homozygosity) produced stronger effects ranging in magnitude between 27.5% and 355.7% above those observed when the variant was present in a single allele (heterozygosity).”
Reviewer 2 Report
Comments and Suggestions for Authors
This is a comprehensive account of potential genetic variations implicated in DNA repair mechanisms related to cerebral and cognitive functionality. The authors present association between single nucleotide polymorphism in DNA repair genes and cognitive measures as well as specific evidence between brain volumes and SNP variants. I think the manuscript is well written, but a little dense. I have a few minor comments that I would like to be addressed.
A better explanation between univariate and multivariate analysis should be included and the main contributions may be summarized
I am curious about what gene-set the authors used (if they used) as the "background" set for their analysis.
The authors stress variability in some BER genes linked to cognitive function and brain structure. What about nucleotide excision repair genes?
Figures are not very easy to address the new finding. Could be expressed in a different way? The legends of the figure should also be transferred below the figure
The conclusions are to be made stronger; they are general in the current version. Some numerical results might be presented and the future directions are to be discussed.
Comments on the Quality of English LanguageMinor editing of English language required
Author Response
R2
This is a comprehensive account of potential genetic variations implicated in DNA repair mechanisms related to cerebral and cognitive functionality. The authors present association between single nucleotide polymorphism in DNA repair genes and cognitive measures as well as specific evidence between brain volumes and SNP variants. I think the manuscript is well written, but a little dense. I have a few minor comments that I would like to be addressed.
Point 1:
A better explanation between univariate and multivariate analysis should be included and the main contributions may be summarized
Response:
As requested we have clarified our description of univariate and multivariate analyses as follows (line 164):
“Univariate models testing each SNP separately were first conducted to determine whether individual SNPs were significantly associated with outcome measures. Next multivariate models including all SNPs with significant associations detected in univariate analyses with one or more cognitive outcome were conducted. The models were progressively reduced (using stepwise deletion) to only include those SNPs that remained significant. This was done to identify the most influential SNPs.”
And (line 319):
“In univariate analyses, which tested each variant individually, all but two SNPs were significantly associated with cognitive performance with five suggestive of a protective effect and 3 of a harmful effect. In contrast, in multi-variate analyses, which tested all variants found to have significant associations in univariate analyses, significant associations after model reduction were only found for eight SNPs with three suggestive of a protective effect and four of a harmful effect.”
Point 2:
I am curious about what gene-set the authors used (if they used) as the "background" set for their analysis.
Response:
We searched the SNPedia library for all documented SNPs located on base excision repair genes which were found to be associated with ageing-related phenotypes (e.g. dementia, cardio-vascular disease, diabetes, etc.) in published research including UNG, TDG, MBD4, ERCC4, OGG1, MUTYH, NTHL1, MGP, NEIL 1-2-3, APEX1, LIG3, XRCC1, PNKP.
Point 3:
The authors stress variability in some BER genes linked to cognitive function and brain structure. What about nucleotide excision repair genes?
Response:
In future studies we plan to investigate nucleotide excision repair genes, but this was beyond the scope of the present investigation which was aimed as a proof of concept. We will use the present evidence base to support a larger project investigating a larger set of DNA repair mechanisms using Mendelian randomisation.
Point 4:
Figures are not very easy to address the new finding. Could be expressed in a different way? The legends of the figure should also be transferred below the figure
Response:
We have used a standard format applied widely across the literature to present the interactions in our figures. In our view this is the clearest way to present them. However, we have now more clearly described them in their respective legends which we have moved as requested below the figures.
Point 5:
The conclusions are to be made stronger; they are general in the current version. Some numerical results might be presented and the future directions are to be discussed.
Response:
As suggested, we have strengthened the conclusions, added some numerical results, and discussed further future directions as follows (page 14):
“The present findings provide convincing evidence that variability in genes implicated in DNA repair, and particularly NEIL1, NEIL2, NTHL1, and OGG1, are associated with cognitive function and brain structure in ageing. These effects are substantial as variants were found to be associated with up to 6% difference in performance of a pair matching task. This suggests that the variants investigated moderate the efficiency of mechanisms coded by BER genes. These differences in efficiency may lead to differential accumulation of DNA damage in brain cells and cause mutations in genes important for brain function, which ultimately could impact individuals’ cognitive function and the underlying brain structure to varying degrees. However, the exact mechanisms involved are unknown and warrant further investigation. These findings are particularly important because the variants investigated are not rare with the most influential ones being present in between 3% and 40% of the population studied. There is therefore a pressing need for large-scale, holistic investigations of the role variability in DNA repair and other related mechanisms contribute to heterogeneous ageing trajectories, as well as functional analysis of variants in DNA repair genes that affect cognition. Future research should also aim to clarify how differences in efficiency of DNA repair interacts with environmental exposures, with a particular focus on modifiable lifestyle and health risk factors as a better understanding of their independent and synergetic effects is likely to provide important insight that may inform prevention, identification of individuals at risk, and the development of new pharmaceuticals.”
Reviewer 3 Report
Comments and Suggestions for Authors
The manuscript “Cognitive function is associated with genetically determined efficiency of DNA repair mechanisms” by Cherbuin et al., explores the links between genetic variations in DNA repair mechanisms, specifically BER-related SNPs, and cognitive function, alongside brain structure. The study highlights the intricate interplay of genetic factors in cognitive aging thereby emphasizing the potential role of DNA repair pathways in shaping cognitive health across diverse populations. Specific comments below.
1. Line 87-90: Authors should elaborate the following sentence briefly to explain what are the evidences or indications of involvement of BER and NER in the brain and cognitive health “Based on the limited existing evidence BER and NER are the repair mechanisms whose genetic variability appears to be most relevant to brain and cognitive health, but we currently do not know how much they contribute to age-related differences in cognitive function or brain health”.
2. In context to the above point authors should also briefly discuss why only SNPs in BER was studied in association with cerebral and cognitive health among other DNA repair mechanisms.
3. Did authors find any correlation of the cognitive function or brain health with different demographics. Information on the demographic characteristics about the study population such as ethnic diversity, socioeconomic status, and any other relevant information could contribute to a more comprehensive understanding of the cohort.
4. Authors may want to explore potential explanations for the observed age and sex-related differences in the associations between the risk index and brain volumes. Consider discussing hormonal influences, neuroinflammatory processes, or other factors that may contribute to these sex-specific patterns.
5. Mention potential mechanisms through which DNA repair mechanisms might act as neuroprotective factors or confer vulnerability to cognitive decline. Discuss existing literature on how BER genes may modulate susceptibility to neurodegenerative conditions.
Author Response
R3
The manuscript “Cognitive function is associated with genetically determined efficiency of DNA repair mechanisms” by Cherbuin et al., explores the links between genetic variations in DNA repair mechanisms, specifically BER-related SNPs, and cognitive function, alongside brain structure. The study highlights the intricate interplay of genetic factors in cognitive aging thereby emphasizing the potential role of DNA repair pathways in shaping cognitive health across diverse populations. Specific comments below.
Point 1:
Line 87-90: Authors should elaborate the following sentence briefly to explain what are the evidences or indications of involvement of BER and NER in the brain and cognitive health “Based on the limited existing evidence BER and NER are the repair mechanisms whose genetic variability appears to be most relevant to brain and cognitive health, but we currently do not know how much they contribute to age-related differences in cognitive function or brain health”.
Response:
As suggested, we have expanded on this point as follows (page 2, lines 87-97):
“Based on the limited existing evidence BER and NER are the repair mechanisms whose genetic variability appears to be most relevant to brain and cognitive health18, 19. This is because the brain is the organ with the highest metabolic rate and as such produces a high amount of reactive oxygen species (ROS)20, 21. As in other parts of the body a large proportion of ROS are buffered by anti-oxidants. However, the remainder may contribute to proportionally higher OS levels and related DNA damage. Indeed, elevated levels of DNA damage in the CNS of animal models have been reported22. Moreover, in post-mitotic cells such as neurons, DNA damage is not mitigated by the more robust DNA repair mechanisms involved in cell replication, but by less reliable DNA repair mechanisms involved in transcription to which BER and NER also contribute23.”
Point 2:
In context to the above point authors should also briefly discuss why only SNPs in BER was studied in association with cerebral and cognitive health among other DNA repair mechanisms.
Response:
To address this point we have added the following to the discussion (page 2, lines 97-108):
“Thus, it would be expected that variability in their efficiency, and particularly BER given it is the major pathway to repair oxidative DNA damage22-24, would lead to more or less damage accumulating in the brain. Such damage accumulation and increased vulnerability to oxidative agents has been demonstrated in cell cultures25, and variability in BER genetic variants was found to be associated with increased DNA damage in Alzheimer dis-ease patients compared to controls26. However, we currently do not know how much such variability contributes to age-related differences in cognitive function or brain health.
The objective of this study is to determine whether genetic variability (SNPs) in the DNA repair mechanism contributing most to resolving OS-related damage (BER) is associated with cerebral, and cognitive health.
Point 3:
Did authors find any correlation of the cognitive function or brain health with different demographics. Information on the demographic characteristics about the study population such as ethnic diversity, socioeconomic status, and any other relevant information could contribute to a more comprehensive understanding of the cohort.
Response:
Generally, minimal correlations were detected between the outcomes measures and covariates. Age, as expected showed the highest correlation with cognitive function, but even these did not reach high levels. We have now added to the manuscript a supplementary Figure S1 which presents these correlations.
Point 4:
Authors may want to explore potential explanations for the observed age and sex-related differences in the associations between the risk index and brain volumes. Consider discussing hormonal influences, neuroinflammatory processes, or other factors that may contribute to these sex-specific patterns.
Response:
Thank you for this suggestion. We have found little plausible pathways explaining this sex and age interaction and do not want to make vague or unsubstantiated hypotheses. Nonetheless, we have identified interesting new evidence that prompted us to add the following (page 14, lines 371-378):
“The origin of this age and sex difference is unclear. A possible explanation is that the three variants, all belonging to NEIL3, which discriminate between class 1 and class 2 somehow interact with female sex hormones. Intriguingly, a recent study37 in an Alzheimer mouse model deficient for NEIL3 showed an age-dependent decrease in amyloid beta plaque deposition in females, which was not observed in males and which was not attributable to increased DNA damage as similar levels were observed in the two sexes. This study did no investigate NEIL3 variant but may suggest that the efficiency in this gene somehow interacts with female sex hormones to influence cognitive function.”
Point 5:
Mention potential mechanisms through which DNA repair mechanisms might act as neuroprotective factors or confer vulnerability to cognitive decline. Discuss existing literature on how BER genes may modulate susceptibility to neurodegenerative conditions.
Response:
The following discussions on possible mechanisms are included in the manuscript (page 2, lines 87-108):
“Based on the limited existing evidence, BER and NER are the repair mechanisms whose genetic variability appears to be most relevant to brain and cognitive health18, 19. This is because the brain is the organ with the highest metabolic rate and as such produces a high amount of reactive oxygen species (ROS)20, 21. As in other parts of the body a large proportion of ROS are buffered by anti-oxidants. However, the remainder may contribute to proportionally higher OS levels and related DNA damage. Indeed, elevated levels of DNA damage in the CNS of animal models have been reported22. Moreover, in post-mitotic cells such as neurons, DNA damage is not mitigated by the more robust DNA repair mechanisms involved in cell replication, but by less reliable DNA repair mechanisms involved in transcription to which BER and NER also contribute23. Thus, it would be expected that variability in their efficiency, and particularly BER given it is the major pathway to repair oxidative DNA damage22-24, would lead to more or less damage accumulating in the brain. Such differences in damage accumulation and increased vulnerability to oxidative agents has been demonstrated in cell cultures25, and variability in BER genetic variants was found to be associated with increased DNA damage in Alzheimer disease patients compared to controls26.”
And (page 14, lines 373-380):
“The origin of this age and sex difference is unclear. A possible explanation is that the three variants, all belonging to NEIL3, which discriminate between class 1 and class 2 some-how interact with female sex hormones. Intriguingly, a recent study37 in an Alzheimer mouse model deficient for NEIL3 showed an age-dependent decrease in amyloid beta plaque deposition in females, which was not observed in males and which was not attributable to increased DNA damage as similar levels were observed in the two sexes. This study did no investigate NEIL3 variant but may suggest that the efficiency in this gene somehow interacts with female sex hormones to influence cognitive function.”